# Simple and Scalable Nearest Neighbor Machine Translation

**Yuhan Dai[‡*], Zhirui Zhang[♯*], Qiuzhi Liu[♯], Qu Cui[♯], Weihua Li[♯], Yichao Du[‡] and Tong Xu[‡]**

[‡]University of Science and Technology of China  [♯]Tencent AI Lab

[‡]`dirkiedye@gmail.com,duyichao@mail.ustc.edu.cn,tongxu@ustc.edu.cn`
[♯]`zrustc11@gmail.com,{qiuzhiliu,qucui,weihuali}@tencent.com`

## ABSTRACT

$k$NN-MT (Khandelwal et al., 2021) is a straightforward yet powerful approach for fast domain adaptation, which directly plugs pre-trained neural machine translation (NMT) models with domain-specific token-level $k$-nearest-neighbor ($k$NN) retrieval to achieve domain adaptation without retraining. Despite being conceptually attractive, $k$NN-MT is burdened with massive storage requirements and high computational complexity since it conducts nearest neighbor searches over the entire reference corpus. In this paper, we propose a simple and scalable nearest neighbor machine translation framework to drastically promote the decoding and storage efficiency of $k$NN-based models while maintaining the translation performance. To this end, we dynamically construct an extremely small datastore for each input via sentence-level retrieval to avoid searching the entire datastore in vanilla $k$NN-MT, based on which we further introduce a distance-aware adapter to adaptively incorporate the $k$NN retrieval results into the pre-trained NMT models. Experiments on machine translation in two general settings, static domain adaptation, and online learning, demonstrate that our proposed approach not only achieves almost 90% speed as the NMT model without performance degradation, but also significantly reduces the storage requirements of $k$NN-MT.

## 1 INTRODUCTION

Domain adaptation is one of the fundamental challenges in machine learning which aspires to cope with the discrepancy across domain distributions and improve the generality of the trained models. It has attracted wide attention in the neural machine translation (NMT) area (Britz et al., 2017; Chen et al., 2017; Chu & Wang, 2018; Bapna & Firat, 2019; Bapna et al., 2019; Wei et al., 2020). Recently, $k$NN-MT and its variants (Khandelwal et al., 2021; Zheng et al., 2021a;b; Wang et al., 2022a) provide a new paradigm and have achieved remarkable performance for fast domain adaptation by retrieval pipelines. These approaches combine traditional NMT models (Bahdanau et al., 2015; Vaswani et al., 2017) with a token-level $k$-nearest-neighbour ($k$NN) retrieval mechanism, allowing it to directly access the domain-specific datastore to improve translation accuracy without fine-tuning the entire model. By virtue of this promising ability, a single $k$NN-MT can be seamlessly generalized to other domains by simply altering the external knowledge it attends to.

In spite of significant achievements and potential benefits, the critical bottleneck of $k$NN-MT is its large token-level external knowledge (also called a datastore), which brings massive storage overhead and high latency during inference. For instance, Khandelwal et al. (2021) found that $k$NN-MT is two orders of magnitude slower than the base NMT system in a generation speed when retrieving 64 keys from a datastore containing billions of records. To ease this drawback and make $k$NN search more efficient, a line of works (Martins et al., 2022b; Wang et al., 2022a) proposed methods to reduce the volume of datastore, such as pruning the redundant records and reducing the dimension of keys. On another line, Meng et al. (2022) designed Fast $k$NN-MT to construct a smaller datastore for each source sentence instead of consulting the entire datastore. Typically, the small datastore is constructed by searching for the nearest token-level neighbors of the source tokens and mapping them to the corresponding target tokens. However, in essence, Fast $k$NN-MT

---

*Equal contribution.

Table 1: Analysis on samples involved in $k$NN retrieval ($k$=8) when translating multi-domain German-to-English datasets (IT, Medical and Law). The left column includes the statistics of full data and correspondent $k$NN-MT performance, while the right column shows the averaged number of sentence pairs (Sents) and target tokens (Tokens), the storage overhead of datastore (Datastore) and correspondent performance of our proposed method (SK-MT) on these samples, respectively.

| Domain | Full | | | | Involved Samples During Inference | | | |
|--------|------|------|-----------|--------|------|-------|-----------|-------|
| | Sents | Tokens | Datastore | $k$NN-MT | Sents | Tokens | Datastore | SK-MT |
| IT | 223k | 3.6M | 6.9G | 45.9 | 7.1 | 249 | 0.46M | 46.3 |
| Medical | 248k | 6.9M | 14.0G | 54.2 | 9.0 | 358 | 0.68M | 57.8 |
| Law | 467k | 19.0M | 37.0G | 61.3 | 14.2 | 730 | 1.5M | 62.7 |

migrates the inefficient $k$NN retrieval from the target side to the source side, resulting in still two times slower decoding speed than the standard NMT model. Despite its failure to speed up, the insights behind Fast $k$NN-MT inspire us that it is avoidable to leverage the entire datastore for nearest neighbor search. In addition to decoding acceleration, little attention has been paid to the storage overhead of $k$NN-MT, which also limits its practicability and promotion on real-world online services. Concretely, $k$NN-MT requires nearly 37GB of disk storage for creating a datastore from 467k law-domain sentence pairs, while the plain text takes less than 100MB, meaning prohibitive storage requirements when applied to a larger corpus. Both factors enlighten us to reflect on the necessity of constructing a complete datastore for $k$NN-MT.

To this end, for each test input, we deeply investigate the involved sentence pairs during the decoding process of $k$NN-MT, and the results are shown in Table 1. We collect the minimum sentence pairs that cover all decoding steps of $k$NN-MT on multi-domain German-to-English datasets, including IT, Medical and Law. Interestingly, we discover that during the whole decoding process, a scarce number of training samples (i.e., less than 16 samples on average) in the domain-specific reference corpus are required for each test sentence, while the storage overhead of the datastore built by these samples is negligible. Moreover, combining the dynamic datastore in $k$NN-MT is capable of achieving superior performance than vanilla $k$NN-MT. This phenomenon provides a new perspective that it is feasible to perform sentence-level retrieval to obtain a small number of similar samples for $k$NN-MT generation.

Following this line, we propose a simple and scalable nearest neighbor machine translation framework (SK-MT) to handle the limitations of $k$NN-MT while keeping its performance. This approach first leverages current efficient text retrieval mechanisms, such as BM25 (Robertson & Zaragoza, 2009), to obtain a small number of reference samples that have high overlaps with the input sentence, and then dynamically construct a tiny datastore by forwarding the samples to the pre-trained model. In this way, we avoid searching through the whole datastore for nearest neighbors and drastically improve the decoding and storage efficiency of $k$NN-MT. Based on that, a distance-aware adapter is introduced to adaptively incorporate the $k$NN retrieval results into the pre-trained NMT models. We conduct experiments in two settings, including static domain adaptation and online learning. Experimental results demonstrate that our proposed approach not only achieves almost 90% speed as the NMT model without performance degradation, but also significantly loosens the storage requirements of $k$NN-MT. Our code is open-sourced on https://github.com/dirkiedai/sk-mt.

## 2 BACKGROUND: $k$NN-MT

Recently, Khandelwal et al. (2021) proposed $k$NN-MT that augments pre-trained NMT models with a translation memory retriever, empowering models to attend to external knowledge and improve translation performance. It is generally formulated in two processes: datastore construction and inference with $k$NN retrieval.

**Datastore Construction.** The datastore is a translation memory which converts bilingual sentence pairs into a set of key-value pairs. Given a reference corpus $(x, y) \in (\mathcal{X}, \mathcal{Y})$, the pre-trained NMT model generates the context representation $f_\theta(x, y_{<t})$ at each timestep $t$. Then we collect the output hidden state $f_\theta(x, y_{<t})$ as key and $y_t$ as value to construct the whole datastore $(\mathcal{K}, \mathcal{V})$:

$$(\mathcal{K}, \mathcal{V}) = \bigcup_{(x,y) \in (\mathcal{X}, \mathcal{Y})} \{(f_\theta(x, y_{<t}), y_t), \forall y_t \in y\}. \tag{1}$$

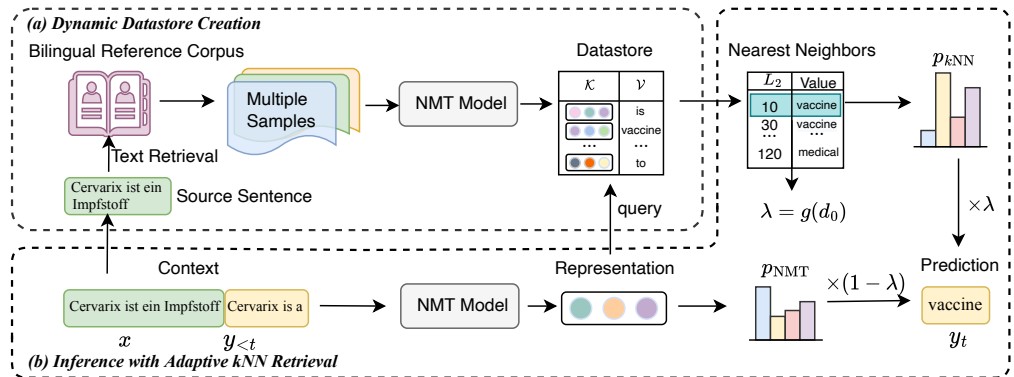

Figure 1: An illustration of our method consisting of two modules, including dynamic datastore construction and inference with adaptive $k$NN retrieval.

**Inference with $k$NN Retrieval.** At the $t$-th decoding step, given the already generated words $\hat{y}_{<t}$, the current context representation $f_\theta(x, \hat{y}_{<t})$ is leveraged to generate a retrieval distribution $p_{k\text{NN}}(y_t|x, \hat{y}_{<t})$ over the entire vocabulary:

$$p_{k\text{NN}}(y_t|x, \hat{y}_{<t}) \propto \sum_{(h_i, v_i) \in N_t} \mathrm{I}_{y_t=v_i} \exp\left(\frac{-d(h_i, f_\theta(x, \hat{y}_{<t}))^2}{\tau}\right), \tag{2}$$

where the $d(.,.)$ stands for Euclidean distance function and $\tau$ is the temperature to control the sharpness of softmax function. The final prediction distribution enhances vanilla NMT distribution $p_{\text{NMT}}$ with the retrieval distribution $p_{k\text{NN}}$, and it is formally calculated as:

$$p(y_t|x, \hat{y}_{<t}) = \lambda p_{k\text{NN}}(y_t|x, \hat{y}_{<t}) + (1-\lambda) p_{\text{NMT}}(y_t|x, \hat{y}_{<t}), \tag{3}$$

where $\lambda$ is a tuned interpolation coefficient. In consideration of beam search in the inference phase, the $k$NN search over the entire datastore $(\mathcal{K}, \mathcal{V})$ is applied at each decoding step of all beams, resulting in the total time complexity of $\mathcal{O}(|\mathcal{K}|Bl)$, where $B$ denotes the beam size and $l$ denotes the target length. Moreover, the storage requirement of datastore leads to the space complexity of $\mathcal{O}(|\mathcal{K}|h)$, where $h$ stands for the dimension of hidden representations. The decisive factor in time and space complexity is the datastore size $|\mathcal{K}|$, therefore the growing of $|\mathcal{K}|$ will engender massive storage overhead and high latency. In practice, $k$NN-MT usually adopts the FAISS (Johnson et al., 2021) toolkit for efficient similarity approximate search and clustering of dense vectors. However, when given an extensive datastore (e.g., billions of records), decoding and storage efficiency is still unsatisfactory (Khandelwal et al., 2021).

## 3 METHODOLOGY

In this work, we provide a new perspective that leverages sentence-level retrieval to dynamically build an extremely small datastore for each test sentence, instead of constructing a complete datastore for $k$NN-MT. It is inspired by an interesting phenomenon that only a few training samples in the domain-specific reference corpus are involved during the decoding process of $k$NN-MT for each test input, as shown in Table 1. We design a simple and scalable nearest neighbor machine translation framework (SK-MT) to handle the shortcoming of $k$NN-MT while keeping its performance in fast domain adaptation. As illustrated in Figure 1, the overall framework can be decomposed into two main processes, i.e., dynamic datastore construction and inference with adaptive $k$NN retrieval. Next, we will give a detailed view of these two modules.

### 3.1 DYNAMIC DATASTORE CONSTRUCTION

In our framework, adopting a dynamic datastore instead of a static and extensive datastore used in conventional $k$NN-MT plays a vital role in fast domain adaptation. Its benefits are mainly two-fold: 1) it filters out the majority of noises in the original datastore in advance and forces the model to attend to the most helpful records, and 2) it limits the nearest neighbor search to an extremely small

datastore, thus improving inference efficiency and reducing storage requirements to a large extent. Specifically, for each test sentence $x$, we leverage an existing search engine (e.g., ElasticSearch[1]) to retrieve the top-64 (default setup) bilingual sentence pairs $\{(x^i, y^i)\}_{i=1}^{64}$ with the highest relevance score from the training corpus. We implement the relevance score as BM25 (Robertson & Zaragoza, 2009), a popular and competitive approach to text ranking tasks. Then in pursuit of selecting a better output hypothesis, we employ the following similarity to re-rank the retrieved bilingual sentences and maintain top-$m$ bilingual reference samples $\{(x^i, y^i)\}_{i=1}^m$:

$$\text{sim}(x, x^i) = 1 - \frac{\text{dist}(x, x^i)}{\max(|x|, |x^i|)}, \tag{4}$$

where $\text{dist}(.,.)$ denotes the edit-distance, and $x^i$, $y^i$ are the retrieved source and target sentences from the training data, respectively. These sentence pairs are utilized to create a small datastore $(\mathcal{K}_x, \mathcal{V}_x)$ for the source sentence $x$ as Equation 1 by forwarding the trimmed corpus to the pretrained NMT model:

$$(\mathcal{K}_x, \mathcal{V}_x) = \bigcup_{1 \le i \le m} \{(f_\theta(x^i, y^i_{<t}), y^i_t), \forall y^i_t \in y^i\}. \tag{5}$$

In this way, the time and space complexity of $k$NN-MT are reduced to $\mathcal{O}(|\mathcal{K}_x|Bl)$ and $\mathcal{O}(|\mathcal{K}_x|h)$ respectively, where $|\mathcal{K}_x| \ll |\mathcal{K}|$.

## 3.2 Inference with Adaptive $k$NN Retrieval

Since we build a significantly small datastore from several similar bilingual sentences, it is inevitable to introduce irrelevant sub-units (chunks or words) for the nearest neighbor search. Under this circumstance, arbitrarily introducing $k$NN distribution with a fixed coefficient $\lambda$ will take the risk of being interfered by the noises in retrieved neighbors. Moreover, it is unnecessary to incorporate $k$NN distribution when the pre-trained NMT model has higher confidence in predicting the next word, otherwise, it tends to harm the performance. To ease the bias of using a fixed interpolation coefficient in $k$NN-MT, prior works (Zheng et al., 2021a) train a light-weight network to predict the coefficient $\lambda$, which decides the engagement of the $k$NN module in an adaptive manner. In this work, we extremely simplify the process by proposing to determine $\lambda$ linear to the normalized distance, which is effective but requires no further training. Precisely, we explicitly calculate $\lambda$ by:

$$\lambda = g(d_0) = \text{ReLU}(1 - \frac{d_0}{\tau}), \tag{6}$$

where $\tau$ is the temperature parameter defined in Equation 2 and $d_0$ represents the top-1 distance during nearest neighbor search. In this way, the engagement of $k$NN distribution is ignored when the distance is larger than the temperature value $\tau$, and it is magnified when records are relevant enough to the current query.

## 3.3 Discussions

To our knowledge, the most relevant work in $k$NN-MT area to our study is Fast $k$NN-MT (Meng et al., 2022), which proposes to construct different datastores for each source sentence by token-level prefiltering. However, it still suffers from constructing the entire datastore in advance, resulting in less decoding and storage efficiency than our SK-MT method that adopts a dynamic datastore. SK-MT can also be viewed as a novel sentence-level pruning strategy, parallel to previous token-level pruning approaches, such as random pruning (Khandelwal et al., 2021), greedy merging (He et al., 2021) and cluster-based pruning (Wang et al., 2022a). Our approach benefits from the fact that the storage overhead of text data is extremely more negligible than dense vectors, and thus we provide a practical version of $k$NN-MT in real applications. Another main advantage of our method is its scalability in terms of reference fetching (owing to the advanced text retrieval system) and translation (owing to the scalability of reference samples). It is effortless to perform operations on a datastore such as insertions, deletions, or substitutions by operating the reference samples, while the same process is quite costly for a $k$NN-based method that requires a pre-defined datastore.

---

[1]https://github.com/elastic/elasticsearch

Interestingly, the retrieval-then-generation paradigm inspired by $k$NN-MT analysis also falls into translation-memory area (Gu et al., 2018; Zhang et al., 2018a; Xia et al., 2019; Cai et al., 2021). Our approach is somewhat in a similar framework of Gu et al. (2018) but introduces $k$NN retrieval to achieve shallow fusion. In this manner, we inherit the advantage of $k$NN-MT that we do not need extra training, and leverage sentence-level text retrieval from the translation-memory area to improve $k$NN-MT's efficiency. We also find that SK-MT achieves better or comparable performance than state-of-the-art translation-memory methods, which are included in the Appendix B.

## 4 EXPERIMENTS

We evaluate the effectiveness of our proposed SK-MT in two settings: 1) domain adaptation, in which a pre-trained general-domain NMT model is used to translate domain-specific sentences with $k$NN searching over an in-domain datastore; 2) online learning from human feedback in the human-in-the-loop machine translation, where the pre-trained NMT model incrementally takes human-corrected samples into account during the decoding process.

### 4.1 DOMAIN ADAPTATION

**Dataset and Evaluation.** For the domain adaptation task, we use the same multi-domain dataset as the baseline (Khandelwal et al., 2021) and consider IT, Medical, Koran, and Law in our experiments. The statistics of the dataset are shown in Appendix A.1. For performance evaluation, we adopt SacreBLEU (Post, 2018)[2] and ChrF (Popovic, 2015).

**Baselines.** We compare our method (**SK-MT**) with several baselines: ① **NMT**: We adopt the WMT'19 German-English news translation task winner model (Ng et al., 2019) as the pre-trained NMT model. ② $k$**NN-MT**: Following Khandelwal et al. (2021), we incorporate the datastore built from domain datasets into the pre-trained model. ③ **AK-MT**: A variant of $k$NN-MT (Zheng et al., 2021a), where the hyper-parameter $k$ is adaptively selected. ④ **FK-MT**: Fast kNN-MT proposed by Meng et al. (2022) constructs a smaller datastore for each source sentence by searching for the nearest token-level neighbors. ⑤ **EK-MT**: An efficient $k$NN-MT proposed by Martins et al. (2022b), which explores several approaches for acceleration. ⑥ **CK-MT**: Chunk-based $k$NN-MT (Martins et al., 2022a) retrieves chunks of tokens from the datastore, instead of a single token.

**Implementation Details.** In our experiments, we adopt the THUMT (Zhang et al., 2020) toolkit for our model implementation. For each sentence in the test set, we retrieve 64 sentences with the highest BM25 scores from the training corpus and perform re-ranking according to the metric defined in Equation 4. We maintain the top-$m$ bilingual sentences as our reference corpus, which are utilized to build a dynamic datastore with the hidden dimension set to 1024. During inference, we carefully tune the hyper-parameters on the development set by performing a grid search on $k \in \{1, 2, 3, 4\}$, $m \in \{1, 2, 4, 8, 16\}$ and $\tau \in \{5, 10, 20, 50, 100, 150, 200\}$. Based on the validation results, we select two widely-used model architectures in our experiments, $m = 2, k = 1$ as SK-MT$_1$ and $m = 16, k = 2$ as SK-MT$_2$, where the temperature $\tau$-s are both set to 100. The beam size and length penalty are set to 4 and 0.6 for all datasets. To replicate other $k$NN-based baselines, we utilize FAISS (Johnson et al., 2021) for efficient $k$NN retrieval, and we learn 4096 cluster centroids for $k$NN retrieval and search 32 clusters for each target token.

Table 4: Grid search on $m$ and $k$ on IT development set with the temperature $\tau$ fixed to 100. The $*$ marks the two selected models (SK-MT$_1$ and SK-MT$_2$) in our experiments.

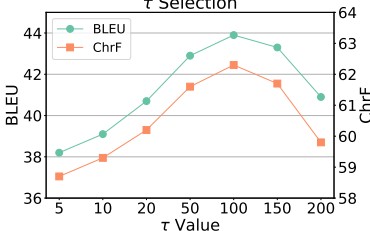

Figure 2: Temperature selection on IT development set.

| $m$ \ $k$ | BLEU | | | | ChrF | | | |
|---|---|---|---|---|---|---|---|---|
| | 1 | 2 | 3 | 4 | 1 | 2 | 3 | 4 |
| 1 | 42.7 | 42.2 | 41.9 | 40.8 | 61.7 | 61.3 | 61.0 | 60.5 |
| 2 | 43.4* | 43.0 | 42.5 | 41.6 | 62.1* | 61.7 | 61.5 | 60.9 |
| 4 | 43.5 | 43.8 | 43.0 | 42.3 | 62.2 | 62.1 | 61.7 | 61.3 |
| 8 | 43.3 | **43.9** | 43.4 | 42.7 | 62.1 | **62.3** | 62.0 | 61.4 |
| 16 | 43.3 | **43.9*** | 43.6 | 43.0 | 62.0 | **62.3*** | 62.1 | 61.8 |

---

[2]https://github.com/mjpost/sacrebleu

Table 2: BLEU($\uparrow$) and ChrF($\uparrow$) on multi-domain test sets, including IT, Medical, Koran and Law.

| Model | BLEU | | | | | ChrF | | | | |
|---|---|---|---|---|---|---|---|---|---|---|
| | IT | Medical | Koran | Law | Avg. | IT | Medical | Koran | Law | Avg. |
| NMT | 39.1 | 41.8 | 16.9 | 45.9 | 35.9 | 58.9 | 61.4 | 39.8 | 66.0 | 56.5 |
| $k$NN-MT | 45.9 | 54.2 | 20.4 | 61.3 | 45.5 | 63.3 | 69.5 | 41.3 | 76.0 | 62.5 |
| AK-MT | **46.9** | **56.4** | 20.3 | **62.6** | **46.6** | **64.4** | **71.0** | **41.7** | **76.9** | **63.5** |
| FK-MT | 45.5 | 53.6 | **21.2** | 56.0 | 44.1 | - | - | - | - | - |
| EK-MT | 44.4 | 51.9 | 20.1 | 57.8 | 43.6 | - | - | - | - | - |
| CK-MT | 44.2 | 53.1 | 19.3 | 59.7 | 44.1 | - | - | - | - | - |
| SK-MT$_1$ | 46.1 | 56.8 | 17.6 | 60.7 | 45.4 | 63.6 | 70.7 | 40.6 | 75.5 | 62.5 |
| - w/o adapter | 40.6 | 47.0 | 18.4 | 52.3 | 39.6 | 59.7 | 63.4 | 40.9 | 69.1 | 58.3 |
| SK-MT$_2$ | **46.2** | **57.6** | 19.5 | **62.3** | **46.4** | **64.0** | **71.3** | 41.5 | **76.4** | **63.3** |
| - w/o adapter | 41.3 | 51.2 | **20.5** | 56.3 | 42.3 | 60.4 | 66.0 | **42.3** | 71.0 | 59.9 |

Table 3: Storage overhead and inference speed on Law test set.

| Model | Storage Overhead | | | Inference Speed (ms/sentence) | | | |
|---|---|---|---|---|---|---|---|
| | Datastore | FAISS Index | GPU | batch=1 | batch=4 | batch=8 | batch=16 |
| NMT | - | - | - | 300.8 ($\times$1.00) | 158.2 ($\times$1.00) | 85.1 ($\times$1.00) | 79.1 ($\times$1.00) |
| $k$NN-MT | 37.0G | 1.3G | ✗ | 1986.4 ($\times$0.15) | 749.5 ($\times$0.21) | 467.4 ($\times$0.18) | 410.5 ($\times$0.19) |
| | | | ✔ | 409.6 ($\times$0.73) | 195.3 ($\times$0.81) | 110.5 ($\times$0.77) | 100.5 ($\times$0.79) |
| SK-MT$_1$ | 0.16M | - | - | 344.4 ($\times$0.87) | 184.8 ($\times$0.86) | 94.6 ($\times$0.90) | 85.0 ($\times$0.93) |
| SK-MT$_2$ | 1.34M | - | - | 430.5 ($\times$0.70) | 225.1 ($\times$0.70) | 131.1 ($\times$0.65) | 136.2 ($\times$0.58) |

**Main Results.** In Table 2, we verify the performance of SK-MT on the multi-domain dataset. As for SK-MT$_1$ that refers to only $m = 2$ samples instead of the whole training set as vanilla $k$NN-MT does, we do not notice significant performance degradation both in terms of BLEU and ChrF. Surprisingly, with a higher amount of reference samples, SK-MT$_2$ substantially outperforms vanilla $k$NN-MT by 1 point of both BLEU and ChrF on average and achieves comparable performance to AK-MT, which is regarded as the state-of-the-art $k$NN-MT method. It is a remarkable outcome that SK-MT$_1$ and SK-MT$_2$ surpass all the efficient methods (FK-MT, EK-MT, and CK-MT) by a large margin, illustrating that our framework is capable of attaining high effectiveness while it is designed for efficient decoding as those three approaches. Additionally, we conduct an ablation study on our adapter with dynamic parameter strategy, which indicates the benefit of adjusting $\lambda$ in an adaptive manner. We also carry out experiments on a more general setting, i.e., WMT'14 news translation task, but find that $k$NN-based methods could not achieve significant improvements. We believe that low sentence-level similarity greatly attributes to the unfavourable performance on the WMT'14 dataset. More results and discussions are included in Appendix C.

**The Effect of Hyper-parameters.** The translation quality of $k$NN-MT is susceptible to its hyper-parameters $\lambda, \tau, k$, which are reduced to $\tau$ and $k$ only in our SK-MT framework. The number of reference samples $m$ is also an essential hyper-parameter that determines the richness of reference samples and the size of the dynamic datastore. It is a straightforward intuition that the increment of $m$ tends to generate better translations but requires more decoding time. Therefore, carefully selecting $m$ is of vital importance because it balances the effectiveness and efficiency of our proposed method. To achieve a better trade-off, we consider 16 as the maximum of $m$. With grid search on the hyper-parameters, as illustrated in Table 4 and Figure 2, we can see that our proposed SK-MT$_2$ achieves the best performance on the IT development set. More details can be found in Appendix A.3. It is worthwhile to notice that SK-MT$_1$ where $k = 1$ and $m = 2$ achieves comparable performance to the best SK-MT$_2$ with conceptually improved efficiency, which will be verified experimentally in the next section.

**Decoding Speed and Storage Overhead.** In this section, we conduct experiments on comparing the inference speed[3] and storage overhead with other baselines. Considering the completeness and fairness of speed comparison, we test the speeds of all the models with various batch sizes, including 1,4,8, and 16. The hardware we use is 112 cores of Intel(R) Xeon(R) Gold 6258R CPU and a single

---

[3]In our experiment, the preliminary time spent on text retrieval of SK-MT is negligible.

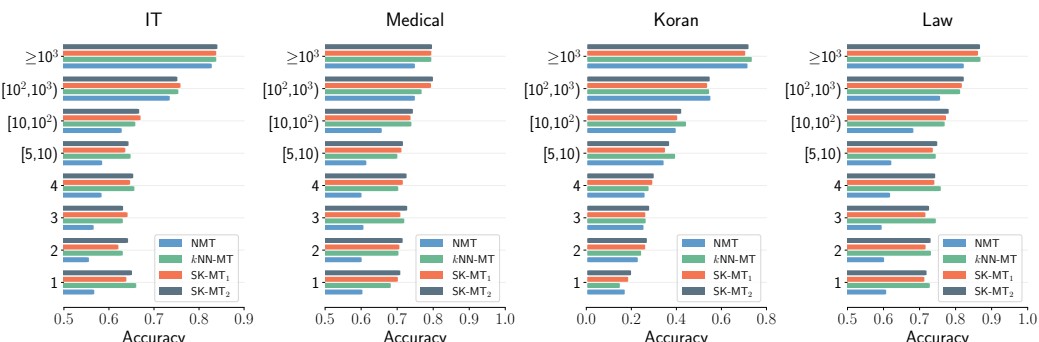

Figure 3: Word accuracy on multi-domain dataset.

GeForce RTX 2080 Ti GPU. As shown in Table 3, our proposed light-weight SK-MT$_1$ achieves almost 90% decoding speed as the NMT model and has higher speed than $k$NN-MT with GPU acceleration. Not surprisingly, the decoding speed of SK-MT will decrease along with the increment of $m$, because typically all the reference samples are passed forward to the NMT models to construct a datastore. In addition, $k$NN-MT typically takes 37G of space to create its datastore, while this storage requirement is avoidable for SK-MT. Both the time and space efficiency of our method are further amplified when given an extensive datastore (e.g., datastore built from WMT'14). More details can be found in Appendix C.

**Word Accuracy Analysis.** It is observed that directly utilizing general models to translate out-of-domain sentences or rare words can cause severe performance degradation (Koehn & Knowles, 2017; Dou et al., 2019). We suspect $k$NN-based models alleviate this issue by introducing external information of out-of-domain and rare words to the pre-trained NMT models. To analyze this problem, we adopt COMPARE-MT (Neubig et al., 2019) to compute the accuracy of words at different frequencies (y-axis). As illustrated in Figure 3, from the perspective of word accuracy, $k$NN-based methods substantially outperform NMT, especially for the rare words at the frequency $\leq 10$. It indicates that the excellent adaptation ability of $k$NN-based models is partially owing to their richer knowledge of low-frequency words. On average, SK-MT$_2$ performs slightly better than SK-MT$_1$, and perhaps it is because the increment of reference samples covers more rare words in the datastore.

## 4.2 ONLINE LEARNING

**Task Description.** We verify the feasibility of applying our proposed approach to machine translation with human feedback, a canonical incremental adaptation task. In this scenario, human experts are involved in post-editing translation hypotheses, and in turn, the revised translations are fed to the MT system for adaptation and improvement. One of the greatest challenges to previous work is that those approaches generally require adapting the online models to new-coming samples constantly, which conduces to significant inefficiency in practice. Worse still, they are suffering from catastrophic forgetting problems. As for our framework, we follow the research line of non-parametric online learning methods (Wang et al., 2022b), which adopt external knowledge to memorize human feedback.

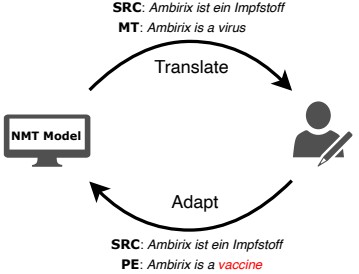

Figure 4: An illustration of the scenario of machine translation with human feedback.

**Dataset and Baselines.** For the online learning task, we adopt two widely-used document-level datasets, i.e., European Medicines Agency (EMEA) dataset (Tiedemann, 2009) and JRCAcquis corpus (Steinberger et al., 2006). Following previous work (Wang et al., 2022b), we divide the documents into five buckets based on their lengths (0-50, 50-100, 100-200, 200-500 and 500-1000). We compare our method with several representative kinds of research, including NMT (Ng et al., 2019), $k$NN-MT (Khandelwal et al., 2021) and KoK (Wang et al., 2022b). More details of dataset and implementation are shown in Appendix A.1 and A.2.

Table 5: BLEU(↑) and ChrF(↑) on EMEA and JRC-Acquis datasets.

| Model | BLEU | | | | | | ChrF | | | | | |
|---|---|---|---|---|---|---|---|---|---|---|---|---|
| | [0, 50) | [50, 100) | [100, 200) | [200, 500) | [500, 1000) | Full | [0, 50) | [50, 100) | [100, 200) | [200, 500) | [500, 1000) | Full |
| EMEA | | | | | | | | | | | | |
| NMT | 44.2 | 43.2 | 38.3 | 42.4 | 40.6 | 41.7 | 64.8 | 63.6 | 61.5 | 63.4 | 64.8 | 64.1 |
| $k$NN-MT | 43.6 | 43.4 | 39.9 | 43.8 | 43.8 | 43.4 | 63.5 | 63.7 | 61.8 | 64 | 65.9 | 64.6 |
| KoK | 44.4 | 44.6 | 44.1 | 45.7 | 53.7 | 49.2 | 65.0 | 65.4 | 64.4 | 65.5 | 71.5 | 68.2 |
| SK-MT$_1$ | 45.5 | 44.8 | 43.4 | 45.6 | 53.2 | 49.2 | 65.3 | 64.7 | 64.3 | 65.5 | 71.6 | 68.3 |
| SK-MT$_2$ | 46.1 | 45.6 | 43.8 | 46.3 | 53.6 | **49.7** | 65.8 | 65.1 | 64.6 | 65.8 | 71.8 | **68.6** |
| JRC-Acquis | | | | | | | | | | | | |
| NMT | 54.1 | 50.0 | 42.2 | 39.9 | 43.4 | 44.5 | 72.2 | 70.2 | 65.9 | 62.9 | 65.6 | 66.4 |
| $k$NN-MT | 55.5 | 52.2 | 45.7 | 43.6 | 47.7 | 48.1 | 72.0 | 70.7 | 67.8 | 65.1 | 68.3 | 68.3 |
| KoK | 56.3 | 52.4 | 47.7 | 44.7 | 50.1 | **49.8** | 73.9 | 72.0 | 69.2 | 66.1 | 70.2 | **69.9** |
| SK-MT$_1$ | 56.6 | 52.8 | 47.2 | 43.5 | 47.8 | 48.5 | 74.0 | 72.0 | 69.0 | 65.3 | 68.8 | 69.1 |
| SK-MT$_2$ | 57.4 | 53.7 | 48.2 | 44.7 | 49.5 | **49.8** | 74.5 | 72.5 | 69.4 | 66.0 | 69.7 | 69.8 |

Figure 5: Results of R-indicator on documents with [50,100), [100,200), [200, 500), and [500,1000) buckets from EMEA and JRC-Acquis.

**Main Results.** We adopt the same performance evaluation as in domain adaptation: BLEU and ChrF. Considering the fairness and completeness of the results, we also measure corpus-level BLEU and ChrF by concatenating the translations of all the documents in each bucket. As illustrated in Table 5, the improvement of KoK over the $k$NN-MT baseline suggests that using a dynamic coefficient $\lambda$ for interpolation helps produce better translation quality, which is also claimed in Wang et al. (2022b). With the benefit of an adaptive $\lambda$, SK-MT$_2$ not only exceeds $k$NN-MT by a large margin as expected, but achieves equivalent or even better performance than KoK on both EMEA and JRC-Acquis datasets. Similar to domain adaptation, SK-MT$_1$ is better than $k$NN-MT in terms of BLEU and ChrF when utilized in online learning setting. Moreover, the improvements in various buckets of length demonstrate the effectiveness and generalization of our method.

**Zero-Shot and Few-Shot Ability.** Following Wang et al. (2022b), we evaluate the adaptation speed of different approaches to human feedback using R-indicator (Simianer et al., 2019), which measures the translation recall of words with different occurrence times in users' feedback. For the $j$-th pair of bilingual pairs of sentences from the corpus $(x_j, y_j) \in (\mathcal{X}, \mathcal{Y})$, we let $\mathcal{R}_{i,j}$ represent unique words in the reference $y_j$ that are their $(i + 1)$-th occurrence in the whole document. We denote $R_i$ as the recall of tokens that have appeared $i$ times in the previous corrected sentences:

$$R_i = \frac{\sum_{j=1}^{|N|} |\mathcal{H}_j \cap \mathcal{R}_{i,j}|}{\sum_{j=1}^{|N|} |\mathcal{R}_{i,j}|} \tag{7}$$

where $N$ denotes the corpus size $|(\mathcal{X}, \mathcal{Y})|$ and $\mathcal{H}_j$ stands for unique words in the $j$-th hypothesis $\hat{y}_j$. Specifically, $R_0$ evaluates the tokens that first appear in the translating document and $R_1$ considers those that have appeared once. We conduct experiments on documents with [50, 100),[100, 200), [200, 500) and [500, 1000) buckets from EMEA and JRC-Acquis datasets, and compute $R_0$, $R_1$, $R_{2\sim5}$, $R_{5\sim9}$ and $R_{9+}$. As shown in Figure 5, in all the settings, SK-MT shows roughly the same trend as KoK, where $R_i$ value of SK-MT improves rapidly and outperforms NMT and $k$NN-MT baselines. It indicates that SK-MT and KoK adapt to helpful human feedback faster.

## 5 RELATED WORK

**Domain Adaptation for NMT.**    Domain adaptation aspires to cope with the discrepancy across domain distributions and improve the generality of the trained models. The domain adaptation approaches in the MT field are mainly divided into two categories: 1) architecture-centric, which typically adds trainable parameters to the NMT model for adaptation. (Wang et al., 2017; Wuebker et al., 2018; Bapna & Firat, 2019; Guo et al., 2021) 2) data-centric, which fine-tunes the NMT model using the domain-specific corpora. In the absence of a pre-defined in-domain corpus, the parallel data can be selected from a larger generic corpus (Del et al., 2021), generated from a monolingual corpus through forward- or back-translation (Zhang et al., 2018b; Poncelas & Way, 2019; Wei et al., 2020), or synthesized from a lexicon or a template (Hu et al., 2019; Peng et al., 2020). Recently, non-parametric methods provide a new paradigm for domain adaptation by retrieving the datastore of similar instances (Khandelwal et al., 2021; Zheng et al., 2021a; Jiang et al., 2021). We follow this research line and propose a more effective and efficient $k$NN-based framework.

**Nearest Neighbor Translation.**    Non-parametric approaches that incorporate external knowledge into the pre-trained models through a retrieval pipeline have attracted wide attention from natural language processing areas, including language modeling (Khandelwal et al., 2020), machine translation (Khandelwal et al., 2021; Zheng et al., 2021a; Jiang et al., 2021), question answering (Guu et al., 2020; Lewis et al., 2020; Xiong et al., 2021) and dialogue generation (Fan et al., 2021; Thulke et al., 2021). For the NMT system, Khandelwal et al. (2021) first propose $k$NN-MT, a non-parametric approach that plugs $k$NN classifier over a large datastore with traditional NMT models (Bahdanau et al., 2015; Vaswani et al., 2017; Hassan et al., 2018) to achieve significant improvement. In addition, Zheng et al. (2021a) propose to dynamically determine the number of retrieved tokens to consider at each step. Martins et al. (2022a) attempt to retrieve chunks of tokens from the datastore, instead of a single token. Due to its scalability and adaptability, $k$NN-MT have also shown great potential for unsupervised domain adaptation (Zheng et al., 2021b; Du et al., 2022) and online learning (Wang et al., 2022b). Despite its great success, $k$NN-MT suffers from large storage overhead and low decoding speed. Recently, several works are proposed to promote its practicality. Meng et al. (2022) design Fast $k$NN-MT to reduce the decoding time, in which they construct different datastores for each source sentence by searching for the neighbours of the source tokens. Martins et al. (2022b) investigate adaptive retrieval, datastore pruning and dimension reduction proposed in He et al. (2021), and further design a retrieval distributions cache to speed-up decoding. Wang et al. (2022a) adopt a lightweight neural network to reduce the dimension of keys and further leverage cluster-based pruning to reduce retrieval redundancy. Different from previous methods, we propose to construct a dynamic datastore with a extremely minor size, which avoids $k$NN search over the entire datastore and thus dramatically improves the time and space efficiency.

## 6 CONCLUSION AND FUTURE WORK

In this paper, we present SK-MT, a simple and scalable nearest neighbor machine translation approach for fast domain adaptation. By constructing a dynamic datastore and introducing a distance-aware adapter for inference, we are able to produce equivalent or even superior performance than previous $k$NN-based approaches. Moreover, experimental results on domain adaptation and online learning settings demonstrate that our framework does not require any extra training and is efficient in both decoding time and storage overhead. It is promising that our proposed SK-MT has a wide range of applications not limited to $k$NN-MT discussed in the paper. In the future, we would like to explore the feasibility of SK-MT when applied in $k$NN-based methods such as $k$NN-LM (Khandelwal et al., 2020), or other sequence-to-sequence tasks.

## ACKNOWLEDGMENTS

We thank the anonymous reviewers for their helpful feedback. We appreciate Lemao Liu and Hongkun Hao for the fruitful discussions and dataset sharing. This work was done when the first author was an intern at Tencent AI Lab and supported by the grants from National Natural Science Foundation of China (No.62222213, 62072423), and the USTC Research Funds of the Double First-Class Initiative (No.YD2150002009). Tong Xu is the corresponding author.

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

## A    EXPERIMENTAL DETAILS AND MORE ANALYSIS

### A.1    DATASET STATISTICS

The statistics of the datasets included in domain adaptation (multi-domain dataset) and online learning (EMEA and JRC-Acquis datasets) tasks are listed in Table 6 and 7, respectively. The Moses toolkit is used to tokenize the sentences and split the words into sub-word units (Sennrich et al., 2016) with the bpe-codes provided by Ng et al. (2019).

Table 6: The statistics of multi-domain dataset.

|  | Koran | IT | Medical | Law |
|---|---|---|---|---|
| Train Sents | 18k | 223k | 248k | 467k |
| Dev Sents | 2000 | 2000 | 2000 | 2000 |
| Test Sents | 2000 | 2000 | 2000 | 2000 |

Table 7: The statistics of EMEA and JRC-Acquis datasets for online learning.

| Bucket | 0-50 | 50-100 | 100-200 | 200-500 | 500-1000 |
|---|---|---|---|---|---|
| | | | EMEA | | |
| Documents | 22 | 14 | 7 | 4 | 5 |
| Ave sentences | 38.4 | 73.0 | 157.9 | 392.8 | 759.2 |
| Ave tokens | 1174.7 | 1938.9 | 3466.1 | 9334.5 | 22725.6 |
| | | | JRC-Acquis | | |
| Documents | 22 | 14 | 7 | 4 | 5 |
| Ave sentences | 38.1 | 73.1 | 158.5 | 373.8 | 734.8 |
| Ave tokens | 1347.1 | 2466.7 | 5345.4 | 12518.2 | 26409.2 |

### A.2    BASELINE SETUP AND IMPLEMENTATION

For the domain adaptation task, we carefully tune the hyper-parameters $\lambda, \tau, k$ of $k$NN-MT (Khandelwal et al., 2021) and report the best scores for each domain. In the implementation of AK-MT (Zheng et al., 2021a), we train a basic model on IT development set about 2500 steps and generalize it to Medical, Koran and Law domains, in which the hyper-parameters $k_{max}, \tau$ are set to 8 and 10, respectively.

As for the online learning task, we consider $k$NN-MT and KoK (Wang et al., 2022b). KoK introduces two $k$-nearest-neighbor ($k$NN) modules: Token-$k$NN memorizes the human feedback, which is the correct sentence provided by human translators, while Policy-$k$NN balances the usage of the history of human feedback and original NMT models adaptively. To replicate KoK, we follow the setup in Wang et al. (2022b) and set the $K$ for Token-$k$NN and Policy-$k$NN to 8. The whole process of the online learning task is as follows: we initialize our reference corpus as empty and incrementally add the corresponding bilingual sentences to the corpus after every source sentence is translated. In this manner, we simulate the human-in-the-loop scenario where the translation system can only attend to the previous human-corrected sentences. When translating the following sentence, we perform the same text retrieval procedure as in the domain adaptation setting for our method. As for $k$NN-MT and KoK, we add the corresponding bilingual sentences to the datastore after every source sentence is translated, which is also a simulation of the online learning scenario.

### A.3    HYPER-PARAMETERS SELECTION

We report the results of adopting a variety of hyper-parameters to our proposed SK-MT model on the development set. We consider grid search on $\tau \in \{5, 10, 20, 50, 100, 150, 200\}$, $k \in \{1, 2, 3, 4\}$ and $m \in \{1, 2, ..., 16\}$. The optimal choices of the multi-domain dataset, including IT, Medical, Koran and Law, are shown in Table 8 and 9.

Table 8: Grid search on temperature $\tau$ with $k$ fixed to 2 and $m$ fixed to 16.

| Domain | Metric | Temperature $\tau$ | | | | | | |
|--------|--------|------|------|------|------|------|------|------|
| | | 5 | 10 | 20 | 50 | 100 | 150 | 200 |
| IT | BLEU | 38.2 | 39.1 | 40.7 | 42.9 | **43.9** | 43.3 | 40.9 |
| | ChrF | 58.7 | 59.3 | 60.2 | 61.6 | **62.3** | 61.7 | 59.8 |
| Medical | BLEU | 45.9 | 48.3 | 51.2 | 54.0 | **55.2** | 54.4 | 52.4 |
| | ChrF | 64.0 | 65.4 | 67.2 | 69.1 | **70.1** | 69.7 | 68.4 |
| Koran | BLEU | 16.6 | 16.5 | 16.8 | 17.6 | 18.2 | **18.5** | **18.5** |
| | ChrF | 39.6 | 39.6 | 39.8 | 40.3 | **40.9** | 40.8 | 40.8 |
| Law | BLEU | 51.0 | 53.0 | 56.1 | 59.7 | **61.3** | 61.1 | 59.8 |
| | ChrF | 68.9 | 70.1 | 72.1 | 74.5 | 75.8 | **75.9** | 75.0 |

Table 9: Grid search on $m$ and $k$ with temperature $\tau$ fixed to 100.

| Domain | $k$ \ $m$ | BLEU | | | | | ChrF | | | | |
|--------|---|------|------|------|------|------|------|------|------|------|------|
| | | 1 | 2 | 4 | 8 | 16 | 1 | 2 | 4 | 8 | 16 |
| IT | 1 | 42.7 | 43.4 | 43.5 | 43.3 | 43.3 | 61.7 | 62.1 | 62.2 | 62.1 | 62.0 |
| | 2 | 42.4 | 43.0 | 43.8 | **43.9** | **43.9** | 61.3 | 61.7 | 62.1 | **62.3** | 62.3 |
| | 3 | 41.9 | 42.5 | 43.0 | 43.4 | 43.6 | 61.0 | 61.5 | 61.7 | 62.0 | 62.1 |
| | 4 | 40.8 | 41.6 | 42.3 | 42.7 | 43.0 | 60.5 | 60.9 | 61.3 | 61.4 | 61.8 |
| Medical | 1 | 53.2 | 54.0 | 54.7 | 54.9 | **55.2** | 68.7 | 69.3 | 69.7 | 69.8 | 69.9 |
| | 2 | 52.1 | 53.5 | 54.6 | 54.8 | **55.2** | 68.0 | 68.8 | 69.6 | 69.7 | **70.1** |
| | 3 | 50.6 | 52.2 | 53.4 | 54.0 | 54.2 | 67.0 | 68.1 | 69.0 | 69.4 | 69.6 |
| | 4 | 49.3 | 51.1 | 52.3 | 53.2 | 53.2 | 66.5 | 67.5 | 68.4 | 69.0 | 69.0 |
| Koran | 1 | 16.3 | 16.9 | 17.2 | 17.4 | 17.4 | 39.4 | 39.8 | 40.1 | 40.2 | 40.2 |
| | 2 | 16.4 | 16.9 | 17.6 | 18.0 | 18.2 | 39.5 | 39.9 | 40.4 | 40.8 | 40.9 |
| | 3 | 16.3 | 16.9 | 17.9 | 18.5 | 18.8 | 39.5 | 39.8 | 40.6 | 41.1 | **41.3** |
| | 4 | 16.2 | 16.6 | 17.9 | 18.6 | **18.9** | 39.5 | 39.7 | 40.5 | 41.1 | 41.1 |
| Law | 1 | 59.6 | 60.6 | 61.0 | 61.1 | **61.6** | 74.7 | 75.3 | 75.6 | 75.7 | **76.0** |
| | 2 | 58.3 | 59.7 | 60.6 | 61.2 | 61.4 | 73.9 | 74.7 | 75.4 | 75.8 | 75.8 |
| | 3 | 57.1 | 58.9 | 59.6 | 60.0 | 60.2 | 73.3 | 74.3 | 74.7 | 75.0 | 75.1 |
| | 4 | 55.8 | 57.6 | 58.3 | 58.9 | 59.4 | 72.6 | 73.6 | 74.0 | 74.5 | 74.6 |

Table 10: BLEU($\uparrow$) and ChrF($\uparrow$) on multi-domain test sets, including IT, Medical, Koran and Law.

| Model | BLEU | | | | | ChrF | | | | |
|-------|------|---------|-------|------|------|------|---------|-------|------|------|
| | IT | Medical | Koran | Law | Avg. | IT | Medical | Koran | Law | Avg. |
| NMT | 39.1 | 41.8 | 16.9 | 45.9 | 35.9 | 58.9 | 61.4 | 39.8 | 66.0 | 56.5 |
| NMT+FT | 48.8 | 57.7 | 21.2 | 62.9 | **47.7** | 66.0 | 71.6 | 42.6 | 77.2 | **64.4** |
| $k$NN-MT | 45.9 | 54.2 | 20.4 | 61.3 | 45.5 | 63.3 | 69.5 | 41.3 | 76.0 | 62.5 |
| AK-MT | 46.9 | 56.4 | 20.3 | 62.6 | **46.6** | 64.4 | 71.0 | 41.7 | 76.9 | **63.5** |
| SK-MT$_1$ | 46.1 | 56.8 | 17.6 | 60.7 | 45.3 | 63.6 | 70.7 | 40.3 | 75.5 | 62.5 |
| SK-MT$_2$ | 46.2 | 57.6 | 19.5 | 62.3 | 46.4 | 64.0 | 71.3 | 41.5 | 76.4 | 63.3 |

## A.4 COMPARISON WITH FINE-TUNED MODELS

Table 10 shows the performance comparisons of $k$NN-based models with the base model fine-tuned on the domain-specific datasets. It can be seen that $k$NN-based methods do not maintain equivalent translation quality to fine-tuned models, underperforming roughly 1 point of BLEU and ChrF on average. However, their domain adaptation performance is acceptable because they do not adapt the pre-trained models for better generation. As for the machine translation with human feedback task, we online update the pre-trained NMT model with human-corrected sentences, and the results are illustrated in Table 11. We can observe that KoK and SK-MT outperform NMT with online tuning in all the buckets of length with no extra constant model updating, which verifies the effectiveness and efficiency of involving non-parametric approaches in the online learning scenario.

Table 11: BLEU($\uparrow$) on EMEA and JRC-Acquis datasets.

| Model | EMEA | | | | | | JRC-Acquis | | | | | |
|---|---|---|---|---|---|---|---|---|---|---|---|---|
| | [0, 50) | [50, 100) | [100, 200) | [200, 500) | [500, 1000) | Full | [0, 50) | [50, 100) | [100, 200) | [200, 500) | [500, 1000) | Full |
| NMT | 44.2 | 43.2 | 38.3 | 42.4 | 40.6 | 41.7 | 54.1 | 50.0 | 42.2 | 39.9 | 43.4 | 44.5 |
| NMT+FT | 44.0 | 43.5 | 39.6 | 43.8 | 44.7 | **43.8** | 54.4 | 50.9 | 43.8 | 42.8 | 47.5 | **47.3** |
| $k$NN-MT | 43.6 | 43.4 | 39.9 | 43.8 | 43.8 | 43.4 | 55.5 | 52.2 | 45.7 | 43.6 | 47.7 | 48.1 |
| KoK | 44.4 | 44.6 | 44.1 | 45.7 | 53.7 | 49.2 | 56.3 | 52.4 | 47.7 | 44.7 | 50.1 | **49.8** |
| SK-MT$_1$ | 45.5 | 44.8 | 43.4 | 45.6 | 53.2 | 49.2 | 56.6 | 52.8 | 47.2 | 43.5 | 47.8 | 48.5 |
| SK-MT$_2$ | 46.1 | 45.6 | 43.8 | 46.3 | 53.6 | **49.7** | 57.4 | 53.7 | 48.2 | 44.7 | 49.5 | **49.8** |

Table 12: The statistics of JRC-Acquis datasets for translation memory methods.

| Dataset | Train Sents | Dev Sents | Test Sents |
|---|---|---|---|
| De$\Leftrightarrow$En | 699,596 | 2454 | 2483 |
| Es$\Leftrightarrow$En | 679,088 | 2533 | 2596 |

## A.5 PERFORMANCE ON DIFFERENT CORPUS SCALES

We further investigate the performance discrepancies brought by different volumes of reference corpus for text retrieval on the Law dataset, and the detailed results are shown in Figure 6. Specifically, we adopt a ratio range of (0.2, 0.4, 0.6, 0.8, 1.0) to randomly sample from the training corpus as our reference corpus for quick experiments, and the same corpus is used to build a datastore for vanilla $k$NN-MT. These results demonstrate that the translation quality improves steadily along with the increment of corpus scale, which verifies the effectiveness of leveraging external information in NMT. SK-MT with $m = 16$ achieves the highest BLEU score among the SK-MT methods and is competitive with the state-of-the-art AK-MT model in corpora with different scales (less than 1 point on average). It is noteworthy to discover that the performance gain between $m = 8$ and $m = 16$ is gradually reduced, meaning that using extensive reference samples when combined with rich external information is unnecessary.

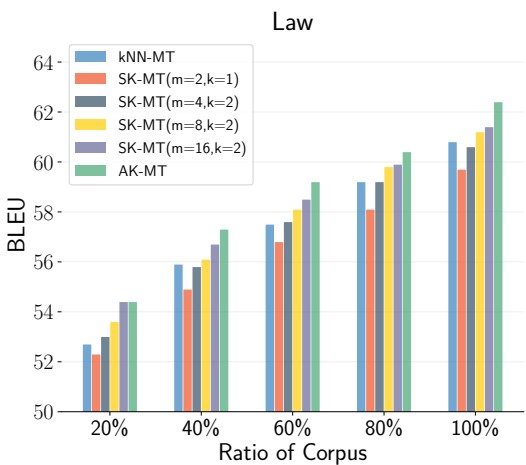

Figure 6: Translation performance of $k$NN-based methods on Law development set given reference corpus of different scales.

## B COMPARISON WITH TRANSLATION-MEMORY METHODS

In order to make a comparison with previous translation memory (TM) approaches (Gu et al., 2018; Zhang et al., 2018a; Xia et al., 2019; Cai et al., 2021), we follow up their experimental setup and evaluate the translation performance on JRC-Acquis corpus (Steinberger et al., 2006). We obtain the datasets originally preprocessed by Gu et al. (2018) and carry out translation experiments on four language pairs, i.e., German-English (De$\Leftrightarrow$En) and Spanish-English (Es$\Leftrightarrow$En). The statistics of datasets are shown in Table 12.

We adopt the same transformer structure as Xia et al. (2019), which contains a 6-layer Transformer encoder and a 6-layer Transformer decoder. The input dimension, FFN layer dimension and attention heads are set to 512, 2048 and 8, respectively. As shown in Table 13, our proposed approach achieves significant improvements or comparable performance to previous TM-based methods, which also indicates the effectiveness of the $k$NN retrieval to achieve translation memory fusion.

Table 13: BLEU score of our method and TM-based approaches on JRC-Acquis corpus.

| Model | Es⇒En dev | Es⇒En test | En⇒Es dev | En⇒Es test | De⇒En dev | De⇒En test | En⇒De dev | En⇒De test |
|---|---|---|---|---|---|---|---|---|
| NMT | 62.8 | 62.7 | 60.4 | 60.5 | 58.5 | 58.9 | 53.3 | 53.3 |
| Gu et al. (2018) | 63.2 | 62.9 | - | - | - | - | - | - |
| Zhang et al. (2018a) | 64.0 | 64.3 | 61.5 | 61.6 | 60.1 | 60.3 | 55.5 | 55.1 |
| Xia et al. (2019) | 66.4 | 66.2 | 62.5 | 62.8 | 61.9 | 61.7 | 57.4 | 56.9 |
| NMT (our implementation) | 64.3 | 64.1 | 62.3 | 61.5 | 59.8 | 60.8 | 55.0 | 54.9 |
| Cai et al. (2021) | **67.0** | 66.5 | 63.0 | 62.8 | 63.6 | 63.9 | 57.9 | 57.5 |
| SK-MT$_1$ | 66.6 | 66.0 | 63.3 | 63.4 | 63.5 | 63.7 | 58.6 | 58.2 |
| SK-MT$_2$ | 66.9 | **66.6** | **64.2** | **63.9** | **63.8** | **64.2** | **59.0** | **59.0** |

Table 14: Model Performance of different methods on WMT'14 test set, where $k$NN-MT adopts FAISS-GPU to speedup the $k$NN retrieval. For inference speed, we set the batch size to 8. The hardware we use is a single NVIDIA Tesla V100 GPU and 96 cores Intel(R) Xeon(R) Platinum 8255C CPU.

| Model | De⇒En BLEU | ChrF | Datastore | FAISS Index | Speed (ms/sent) | En⇒De BLEU | ChrF | Datastore | FAISS Index | Speed (ms/sent) |
|---|---|---|---|---|---|---|---|---|---|---|
| NMT | 31.4 | 58.1 | - | - | 36.7 | 27.2 | 57.8 | - | - | 32.8 |
| $k$NN-MT | 31.3 | 58.0 | 145G | 9.0G | 252.4 | 27.3 | 57.8 | 128G | 7.9G | 343.2 |
| SK-MT$_1$ | 31.4 | 58.0 | 0.06M | - | 50.1 | 27.0 | 57.8 | 0.06M | - | 43.3 |
| SK-MT$_2$ | 31.3 | 58.0 | 0.46M | - | 63.4 | 27.0 | 57.8 | 0.46M | - | 50.4 |

Table 15: Statistics for sentence similarities between each test sentence and the training set for the WMT'14 De⇔En (3003 sentences), JRC-Acquis De⇔En (2483 sentences) and IT De⇒En (2000 sentences) datasets.

| Similarity | WMT'14 De⇒En Sent | Percent | En⇒De Sent | Percent | JRC-Acquis De⇒En Sent | Percent | En⇒De Sent | Percent | IT De⇒En Sent | Percent |
|---|---|---|---|---|---|---|---|---|---|---|
| $[0, 0.1)$ | 1 | 0% | 0 | 0% | 2 | 0% | 0 | 0% | 164 | 8.2% |
| $[0.1, 0.2)$ | 70 | 2.3% | 61 | 2% | 116 | 4.7% | 70 | 2.8% | 111 | 5.6% |
| $[0.2, 0.3)$ | 1210 | 40.3% | 1101 | 36.7% | 344 | 13.6% | 288 | 11.4% | 174 | 8.7% |
| $[0.3, 0.4)$ | 1096 | 36.5% | 1125 | 37.5% | 346 | 13.7% | 353 | 14.1% | 229 | 11.5% |
| $[0.4, 0.5)$ | 411 | 13.7% | 468 | 15.6% | 226 | 0.9% | 225 | 8.9% | 141 | 7.1% |
| $[0.5, 0.6)$ | 173 | 5.8% | 198 | 6.6% | 233 | 9.1% | 246 | 9.7% | 543 | 27.2% |
| $[0.6, 0.7)$ | 28 | 0.9% | 35 | 1.2% | 223 | 8.7% | 213 | 8.4% | 296 | 14.8% |
| $[0.7, 0.8)$ | 9 | 0.3% | 11 | 0.4% | 216 | 0.5% | 228 | 9% | 151 | 7.6% |
| $[0.8, 0.9)$ | 4 | 0.1% | 4 | 0.1% | 369 | 14.8% | 418 | 16.8% | 148 | 7.4% |
| $[0.9, 1]$ | 1 | 0% | 1 | 0% | 441 | 16.6% | 442 | 17.7% | 43 | 2.2% |

## C EXPERIMENTS ON WMT'14 DATASET

In this section, we carry out experiments on the WMT'14 English-German dataset consisting of 4.5M sentence pairs for model training and consider bidirectional translation in this setting. We learn joint bpe-codes at the length of 45K types and adopt the Moses toolkit to tokenize the sentences and split the words into sub-word units. For each translation direction, we train a separate Transformer-based model, and select the best model on the development set, a split of the entire sentence pairs with a ratio of 1%. We adopt SacreBLEU (Post, 2018) and ChrF (Popovic, 2015) for performance evaluation and report final results on newstest2014, which is composed of 3003 bilingual sentence pairs.

As listed in Table 14, we find that $k$NN-based methods show little power in boosting the translation quality when utilized in the WMT'14 translation task. We suspect that it is partially because the pre-trained model is trained on the same reference corpus, which dramatically limits the richness of external information. Regardless, these results prove that SK-MT has much smaller latency than vanilla $k$NN-MT if the datastore is considerable, which reveals that constructing a dynamic datastore

is very helpful in promoting $k$NN-MT's efficiency. Furthermore, to give a more profound insight into this phenomenon, following Zhang et al. (2018a), we measure the similarity between a test sentence $x$ and the training corpus $\mathcal{D}_{train}$ by computing the sentence similarities between $x$ and the retrieved source sentences as

$$\text{sim}(x, \mathcal{D}_{train}) = \max_{x_i \in \mathcal{D}_{train}} \text{sim}(x, x_i) \tag{8}$$

The analysis listed in Table 15 demonstrates that most of the test sentences (nearly 75%) have similarities of less than 0.4 to the training set, suggesting the similarity between training and test sets in the WMT'14 is considerably low. We believe the low similarity greatly attributes to the unfavourable performance of the WMT'14 dataset. And also, the different distribution of similarities between the WMT'14, JRC-Acquis and IT datasets can be used to explain the performance discrepancy listed in Table 2, 13 and 14. Thus, high similarity contributes much to $k$NN-based methods, which are similar to TM-based methods, and it also gives some insights into the working mechanism of $k$NN-based methods.

