# OpenReview forum: "Simple and Scalable Nearest Neighbor Machine Translation"
_ICLR.cc/2023/Conference — ICLR 2023 poster_

### Official Review · Reviewer_XTZD · 2022-10-17

**Confidence:** 5
**Clarity, Quality, Novelty And Reproducibility:** See strengths and weaknesses
**Correctness:** 3
**Technical Novelty And Significance:** 3
**Empirical Novelty And Significance:** 2
**Recommendation:** 6

**Strength And Weaknesses:**

**Strength**
* The method is simple yet effective on multiple adaptation tasks.
* The writing is clear and the paper is easy to follow.

**Weaknesses**
* $k$NN-MT is not only good at (narrow-)domain adaptation, but also good at improving performance on large-scale open-domain task like WMT en-de, however there is no corresponding experiments in this paper. Actually, I suspect that the sentence-level retrieval may only work well on narrow-domain data as reported in [2].
* Regarding the two major contribution of this paper, i.e. 1) search similar sentences by search engine and 2) distance-aware adaptor, there are existed works [1][2][3]. I would give authors the credit of combining of them together in $k$NN-MT setting, but I think the contribution may not be significant enough. Directly comparing the proposed method with [1] and [2] and siginificant improvements could address my concern.
* Although the compuation cost of searching $k$NN is alleviated, the proposed method still need to compute hidden representation of $k$ nearest sentences on-the-fly, this is actually another heavy compuational burden on GPU.

**References**

[1] Search engine guided neural machine translation. Jiatao Gu et al., AAAI 2018

[2] Guiding Neural Machine Translation with Retrieved Translation Pieces. Zhang et al., NAACL 2018

[3] Adaptive nearest neighbor machine translation. Zheng et al., ACL2021

**Summary Of The Paper:**

$k$NN-NMT is a straightforward yet powerful approach for fast domain adaptation, but is burdened with massive storage requirements
and high computational complexity. This paper propose an efficient variant of $k$NN-MT that drastically increase speed in two folds:
1) search similar sentences by search engine instead of searching over all tokens in training datasets
2) use a distance-aware adaptor to adaptively incorporate the kNN retrieval results into the pre-trained NMT models.



**Summary Of The Review:**

Overall, the experimental results on adaptaion tasks seems good, but lack of experiments on large-scale open-domain dataset like WMT.

---

> ### Author Response · Authors · 2022-11-19
> **Response to Reviewer XTZD**
>
> We would like to thank you very much for the detailed feedback and valuable suggestions! We will follow the suggestions on experiments and writing and revise our paper accordingly.
>
> **Q1: Experiment on the WMT14 dataset and comparison with TM-based methods**
>
> A1: We cannot reproduce the results of the original kNN-MT on the WMT19 dataset due to the limited time and hardware resources. We find that the datastore built from the WMT19 dataset reaches TB-level storage. It requires more than 50GB faiss-index to perform kNN approximate search, which means that we cannot load the full faiss-index in one V100/A100 GPU. Worse still, the authors do not release their official codes to replicate kNN-MT on WMT'19.
>
> Instead, we build a datastore on the WMT14 dataset and tested kNN-based approaches on newstest14. Nevertheless, we observe a slight performance improvement on the newstest14, not as significant as what they reported in [1]. We suspect that it is partially because the pre-trained model is trained on the same reference corpus, which dramatically limits the richness of external information. Our method still achieves comparable performance to kNN-MT with significantly less time and space cost if the datastore is considerable, which reveals that constructing a dynamic datastore is very helpful in promoting kNN-MT's efficiency.
>
> To further verify the effectiveness of our proposed SK-MT, we also conducted experiments on the JRC-Acquis dataset. The results demonstrate that SK-MT significantly outperforms other TM-based baselines [2,3,4,5]. At last, we further analyze the different performances of these two datasets following [3]. We measure the similarity between train and test datasets on WMT14, JRC-Acquis, and IT-domain datasets, respectively, and discover that high similarity contributes much to kNN-based methods, which are similar to TM-based methods. **It seems to give some insights into the working mechanism of kNN-based methods. We would add more analysis in the future version.**
>
> Please note all the detailed results and analysis are included in Appendix A.6 and A.7 of our new draft.
>
> **Q2: Computational cost of dynamic datastore**
>
> A2: We have included the computational cost of SK-MT's dynamic datastore construction in Table 3 when evaluating the inference speed. Actually, the dynamic datastore construction is fast, since we only need one force-decoding process over retrieval sentence pairs (this computation is well parallelized) before the beam search.
>
> References:
>
> [1] Urvashi Khandelwal, Angela Fan, Dan Jurafsky, Luke Zettlemoyer, and Mike Lewis. Nearest neighbor machine translation. In ICLR, 2021.
>
> [2] Jiatao Gu, Yong Wang, Kyunghyun Cho, and Victor O. K. Li. Search engine guided neural machine translation. In AAAI, 2018.
>
> [3] Jingyi Zhang, Masao Utiyama, Eiichiro Sumita, Graham Neubig, and Satoshi Nakamura. Guiding neural machine translation with retrieved translation pieces. In NAACL, 2018.
>
> [4] M. Xia, Guoping Huang, Lemao Liu, and Shuming Shi. Graph based translation memory for neural machine translation. In AAAI, 2019.
>
> [5] Deng Cai, Yan Wang, Huayang Li, Wai Lam, and Lemao Liu. Neural machine translation with monolingual translation memory. In ACL, 2021.

---

> > ### Comment · Reviewer_XTZD · 2022-11-29
> > **Thanks for the additional experiments**
> >
> > The improvements over JRC-Acquis addresses my biggest concern, and I'll raise my score to 6. It's a pitty that there is no experiments on WMT19 due to limit of hardware resources. Actually I successfully reproduced kNN-NMT on WMT19 using CPU, maybe you can use CPU instead of GPU to reproduce kNN-NMT.

---

### Official Review · Reviewer_TZ41 · 2022-10-24

**Confidence:** 4
**Correctness:** 3
**Technical Novelty And Significance:** 2
**Empirical Novelty And Significance:** 2
**Recommendation:** 8

**Clarity, Quality, Novelty And Reproducibility:**

This paper is easy to follow and the proposed method is very clear. The proposed method will be easily reproduced given the simplicity of the proposed approach.

**Strength And Weaknesses:**

Strength

* The use of data retrieval based on BM25 sounds a simple yet efficient method to solve the computational problem in kNN-MT. The experiments in Table 3 shows reduced latencies by extracting only relevant bilingual sentences without loss in translation qualities as shown in Table 2.

* The dynamic adaptation for the coefficient in Equation 6 is very simple and experiments demonstrate the gains.

Weakness

* This work presents experiments in two tasks, domain adaptation and online learning, and does not present any results regarding translation in general, e.g., training data on news domain and testing on news domain. This work should also include those results to discuss whether the proposed method also works in a standard task setting. Even if the results were negative, they should be provided for further discussion.

  - The additional experimental results comparing for generic dataset sounds reasonable to me.

* The proposed dynamic coefficient scaling seems to be very effective in adjusting the tradeoff of the extracted dataset, but it is not clear whether the approach is only effective to the BM25-based filtering or not. This work should compare the use of the dynamic scaling for other baselines, e.g., the original kNN MT and its variations, to investigate whether the approach is orthogonal to the retrieval methods or not.

  - I also like the additional experiment, and that should be included in the main body.

**Summary Of The Paper:**

This work proposes two very simple methods to improve the efficiency and effectiveness of k-nearest neighbor machine translation by 1) dynamically creating a datastore of bilingual sentences given an input sentence using BM25, and 2) dynamically adjusting a coefficient to scale the contribution from the similarity scores of extracted instances.

Experiments are carried out in two scenarios, MT domain adaptation and online learning. In the domain adaptation task, the proposed work achieved significant speed up without loosing the MT qualities. In the online learning task, e.g., iteratively enlarging the training data, the proposed method also achieved better results when compared with prior baselines.

**Summary Of The Review:**

This work presents an interesting and very simple approach to solve the efficiency problem in kNN MT. Experiments are well designed, but it is missing an important question on how well the proposed approach might work on a standard setting, i.e., training and testing on the same domain with large data. Also, the dynamic scaling sounds a nice contribution to the field, but it is not clear whether the approach could be employed in other retrieval methods or not.

- Additional experiments will be useful for this community.

---

> ### Author Response · Authors · 2022-11-19
> **Response to Reviewer TZ41**
>
> We would like to thank you very much for the detailed feedback and valuable suggestions! We will follow the suggestions on experiments and writing and revise our paper accordingly.
>
> **Q1: Experiment on WMT14 dataset**
>
> A1: We cannot reproduce the results of the original kNN-MT on the WMT19 dataset due to the limited time and hardware resources. We find that the datastore built from the WMT19 dataset reaches TB-level storage. It requires more than 50GB faiss-index to perform kNN approximate search, which means that we cannot load the full faiss-index in one V100/A100 GPU. Worse still, the authors do not release their official codes to replicate kNN-MT on WMT'19.
>
> Instead, we build a datastore on the WMT14 dataset and tested kNN-based approaches on newstest14. Nevertheless, we observe a slight performance improvement on the newstest14, not as significant as what they reported in [1]. We suspect that it is partially because the pre-trained model is trained on the same reference corpus, which dramatically limits the richness of external information. Our method still achieves comparable performance to kNN-MT with significantly less time and space cost if the datastore is considerable, which reveals that constructing a dynamic datastore is very helpful in promoting kNN-MT's efficiency.
>
> To further verify the effectiveness of our proposed SK-MT, we also conducted experiments on the JRC-Acquis dataset. The results demonstrate that SK-MT significantly outperforms other TM-based baselines [2,3,4,5]. At last, we further analyze the different performances of these two datasets following [3]. We measure the similarity between train and test datasets on WMT14, JRC-Acquis, and IT-domain datasets, respectively, and discover that high similarity contributes much to kNN-based methods, which are similar to TM-based methods. **It seems to give some insights into the working mechanism of kNN-based methods. We will add more analysis in the future version.**
>
> Please note all the detailed results and analysis are included in Appendix A.6 and A.7 of our new draft.
>
> **Q2: Combination of kNN-MT and our proposed adapter**
>
> A2: We conduct an ablation study on combining kNN-MT with our proposed dynamic coefficient scaling method. The results are listed below:
> |    | IT | Medical | Koran | Law |
> |  ----  | ----  |   ----  | ----  | ----|
> kNN-MT | 45.9  | 54.2  | 20.4 | 61.3 |
> kNN-MT w/ adapter | 45.8  | 52.6  | 20.8 | 57.8 |
>
> In the domain adaptation task, our dynamic coefficient does not contribute to performance improvement, and especially the performances on Medical and Law are terrible.
> We argue that it is because the entire datastore contains much noise; introducing a simple scaling method does not have enough ability to filter out the noise. As can be seen in our main experiments as well as the results in Koran, the approach will benefit kNN-MT when its datastore is relatively small. Perhaps this approach is not orthogonal to the retrieval methods.
>
> References:
>
> [1] Urvashi Khandelwal, Angela Fan, Dan Jurafsky, Luke Zettlemoyer, and Mike Lewis. Nearest neighbor machine translation. In ICLR, 2021.
>
> [2] Jiatao Gu, Yong Wang, Kyunghyun Cho, and Victor O. K. Li. Search engine guided neural machine translation. In AAAI, 2018.
>
> [3] Jingyi Zhang, Masao Utiyama, Eiichiro Sumita, Graham Neubig, and Satoshi Nakamura. Guiding neural machine translation with retrieved translation pieces. In NAACL, 2018.
>
> [4] M. Xia, Guoping Huang, Lemao Liu, and Shuming Shi. Graph based translation memory for neural machine translation. In AAAI, 2019.
>
> [5] Deng Cai, Yan Wang, Huayang Li, Wai Lam, and Lemao Liu. Neural machine translation with monolingual translation memory. In ACL, 2021.

---

> > ### Comment · Reviewer_TZ41 · 2022-11-22
> > **Thanks for your comment**
> >
> > I really appreciate your experiments on WMT14 and JRC-Acquis. The analysis will be of help for further investigation. Also, I like the experiment of the adaptive method for the kNN-MT baseline, although the proposed scaling might not be helpful for the baseline.

---

> > > ### Author Response · Authors · 2022-11-29
> > > **Thanks for your reply**
> > >
> > > Thank you very much for your kind reply. Due to the limited time, we were only able to append our additional experiments to the supplementary content. In the future version, we will follow your suggestions move several important points into the main body of our paper.

---

### Official Review · Reviewer_n9iv · 2022-10-24

**Confidence:** 5
**Correctness:** 3
**Technical Novelty And Significance:** 2
**Empirical Novelty And Significance:** 3
**Recommendation:** 6

**Clarity, Quality, Novelty And Reproducibility:**

The paper could improve on clarity. See suggestions in the previous section. Also add descriptions of the baselines to at least the appendix (or main paper if there is space).

The experiments and analysis are interesting, even given the connection to previous work which hurts novelty a bit. But, the framing and discussion are not clear or comprehensive which currently hurts the quality of the work.

There is no mention of code release, but the paper includes details on hyperparameters and setup that would probably help reproducibility, at least on the domain adaptation experiments.


**Strength And Weaknesses:**

Strengths:
1. This paper brings the approach from Gu et al. (2018) to modern transformers and kNN-MT style approaches that don’t require training the model, while having much lower storage costs for the datastore which makes brute-force search faster.
2. It proposes a simplification to adaptive kNN-MT that does not require training a separate network. Ablations show that the proposed adaptation is especially important for the dynamically constructed datastores.
3. The proposed approach is on par with the full datastore kNN-MT and variants.
4. The paper provides latency scores as well as an analysis of word accuracy across methods based on the frequency of the words.
5. The paper provides an analysis of the rate at which various approaches adapt to domain-specific words.

Weaknesses:
1. The paper’s primary proposed approach for building the index dynamically seems very similar to Gu et al. (2018) and specifically their shallow fusion variant. Yet, the paper doesn’t really explore this connection or even attribute this approach to them. The results and analyses are interesting, but a discussion on this connection seems important and is currently missing.

2. While the decrease in storage costs is nice, the decoding speed improvements aren’t clearly there due to faiss-gpu, especially when the number of retrieved translation pairs increases from 2 to 16. The current story and discussion should be revised to remove the claim that this approach improves decoding speed, or it should be revised to clearly state that a trade-off exists – better decoding speed comes at the cost of lower performance. Adding a plot on latency vs. performance would go nicely with a discussion on the trade-off.

3. The paper calls this approach scalable but it is unclear how the number of retrieved translation pairs (m) scales with the size of the corpus. Is this claim justified somewhere in the paper through trends based on corpus size? The original kNN-MT paper shows that larger datastores provide diminishing returns so perhaps there is some evidence of this in the literature but missed the discussion in the paper.

4. The discussion on online learning experiments needs more detail. It’s not very clear what the setup is for the different approaches, especially SK-MT. Is there some constraint that selects human provided translations, or are the token-level retrievals only over the human provided translations?

5. Related to the previous discussion, the analysis on R-values is also a bit confusing. Given the current explanation of the online learning experiments, shouldn’t the R-value for all kNN approaches converge to the same value for tokens that have appeared 9+ times? Why do the full datastore retrievals ignore the human feedback but SK-MT doesn’t, or is something else going on? Also how important is it to upweight the retrievals from human feedback? What if there were a completely separate kNN function operating on the human feedback combined with kNN over the dataset and the base MT system?

6. Overall, given the connection to Gu et al. (2018) and the lack of decoding speed improvements, it isn’t clear if the approach makes enough of a case for added value even though the results and analysis are interesting – at least not in the current narrative of the paper. Why is this approach better than just adaptive kNN-MT if it doesn’t run faster? Both sets of experiments show similar performance to full datastore but adaptive versions of kNN-MT. Maybe one argument is that lower storage makes this viable for running kNN-MT on-device where there are storage constraints?

Questions:
1. Why are numbers in Tables 2 and 4 for IT different?

Jiatao Gu, Yong Wang, Kyunghyun Cho, and Victor O. K. Li. Search engine guided neural machine translation. In AAAI, 2018.


**Summary Of The Paper:**

This paper presents SK-MT which makes storage efficiency improvements over the kNN-MT model. The efficiency comes in the form of a dynamic datastore construction strategy that uses token-level retrieval (BM25) to first get a set of relevant translation pairs, and then construct a datastore from this smaller subset. To alleviate noise from a smaller datastore, an adaptive interpolation parameter is computed based on the ratio of the top-1 neighbor’s distance and the softmax temperature used with kNN-MT. This is a simpler adaptation approach than adaptive kNN-MT that involves learning a separate network for adaptation.

Experiments are conducted on two MT tasks, reporting BLEU and character n-gram f-scores.
- Domain adaptation using the multi-domains data: the proposed SK-MT models are on par with the full-datastore baselines (kNN-MT and adaptive kNN-MT). The proposed SK-MT method reduces storage size of the datastore considerably and eliminates the need for a faiss index, but doesn’t necessarily run faster than the full datastore baselines when they search the faiss index on the GPU.
- MT with humans in the loop providing post-edits and corrections: this involves growing the datastores dynamically as the human edits become available. The two corpora include EMA and JRCAquis. The proposed SK-MT method is on par with the KoK baseline that operates over the full datastore with an adaptive interpolation parameter.


**Summary Of The Review:**

Recommending rejection because the current version of the paper is missing an important connection to prior work and a number of claims are unclear and/or unjustified. Open to revising the recommendation after the discussion period.

EDIT after author response:
The authors did a good job of addressing the primary concerns raised in the initial review. While the novelty of their approach is still limited, this paper makes contributions on the empirical side and provides data points that could be valuable for the community. For this reason, I’m raising my score to a 6.

---

> ### Author Response · Authors · 2022-11-19
> **Response to Reviewer n9iv (Part 1)**
>
>
> We would like to thank you very much for the detailed feedback and valuable suggestions! We will follow the suggestions on experiments and writing and revise our paper accordingly.
>
> **Q1: Discussions and connections with Gu et al. (2018) or other TM-based methods**
>
> A1: Interestingly, the retrieval-then-generation paradigm inspired by kNN-MT analysis also falls into the translation memory area [1,2,3,4]. We agree that our approach is somewhat in a similar framework of [1] but introduces kNN retrieval to achieve shallow fusion. In this manner, we inherit the advantage of kNN-MT that we do not need extra training and leverage sentence-level text retrieval from the translation memory area to improve kNN-MT's efficiency. We also show SK-MT's superiority compared with previous translation memory methods in detailed experiments, which is included in the Appendix A.6 of our new draft. As shown in Table 13, our proposed approach achieves significant improvements or comparable performance to previous TM-based methods, which indicates the effectiveness of the kNN retrieval to achieve translation memory fusion. We would add more discussions with TM-based methods to the related work in the future version.
>
> **Q2: Decoding speed improvement**
>
> A2: We further carry out experiments on the WMT'14 English-German dataset consisting of 4.5M sentence pairs. The details and results are included in Appendix A.7 of our new draft. As shown in Table 14, SK-MT has an extremely smaller latency than vanilla kNN-MT when given an extensive datastore, which reveals that constructing a dynamic datastore is very helpful in promoting kNN-MT's efficiency.
>
> **Q3: Performance on different corpus scales**
>
> A3: We have discussed the scalability of our method in Section 3.3. We claim that our method can scale with corpora with different volumes, especially if they are extensive. In order to investigate how the number of retrieved translation pairs (m) scales with the size of the corpus, we further conduct experiments on the performance discrepancies brought by different volumes of reference corpus for text retrieval on the Law dataset, and the detailed results are included in Appendix A.5 of our new draft. As shown in Figure 6, these results demonstrate that the translation quality improves steadily along with the increment of corpus scale, which verifies the effectiveness of leveraging external information in NMT. SK-MT with m=16 achieves the highest BLEU score among the SK-MT methods and is competitive with the state-of-the-art AK-MT model in corpora with different scales (less than 1 point on average). It is noteworthy to discover that the performance gain between m=8 and m=16 is gradually reduced, meaning that using extensive reference samples when combined with rich external information is unnecessary.
>
> **Q4: More details of the online learning setting**
>
> A4: The whole process of SK-MT in the online learning task is as follows: we initialize our reference corpus as empty and incrementally add the corresponding bilingual sentences to the corpus after every source sentence is translated. In this manner, we simulate the human-in-the-loop scenario where the translation system can only attend to the previous human-corrected sentences. When translating the following sentence, we perform the same text retrieval procedure as in the domain adaptation setting for our method. As for kNN-MT and KoK, they add the corresponding bilingual sentences to the datastore, which is also a simulation of the online learning scenario.
>
> **Q5: R-indicator, KoK and SK-MT in the online learning setting**
>
> A5: R-indicator measures the translation recall of words with different occurrence times in users' feedback. Under this simulated human-in-the-loop scenario, it can be used to evaluate how fast the model can learn from their feedback to avoid repeatedly making the same mistakes they already corrected before. Note that the adaptation ability of each method varies, in which R9+ does not converge to a specific value. Actually, KoK introduces another kNN module to upweight the retrievals from human feedback. But we achieve this by our proposed dynamic coefficient scaling, and our SK-MT framework is competitive with KoK in practice.

---

> ### Author Response · Authors · 2022-11-19
> **Response to Reviewer n9iv (Part 2)**
>
> **Q6: Comparsion with the adaptive kNN-MT (AK-MT)**
>
> A6: Note that we dynamically construct an extremely small datastore for each input via sentence-level retrieval to avoid searching the entire datastore in vanilla kNN-MT, based on which we further introduce a distance-aware adapter to adaptively incorporate the kNN retrieval results into the pre-trained NMT models. These two processes help to develop a practical kNN-based framework while surpassing kNN-MT and even achieving comparable performance to SOTA (AK-MT). We do not claim that approach is better than adaptive kNN-MT in terms of translation quality, but the approach is much more practical than vanilla kNN-based methods (kNN-MT and AK-MT). The time and space efficiency on the WMT'14 dataset are more significant than that in the multi-domain dataset. Please note that all the above contents will be included in our next draft.
>
> **Q7: Why are numbers in Tables 2 and 4 for IT different? Reproducibility?**
>
> A7: Table 2 lists our main results on the test set, and Table 4 contains the hyper-parameter selection results, which are conducted on the development set. We will release our codes after the rebuttal period.
>
> References:
>
> [1] Jiatao Gu, Yong Wang, Kyunghyun Cho, and Victor O. K. Li. Search engine guided neural machine translation. In AAAI, 2018.
>
> [2] Jingyi Zhang, Masao Utiyama, Eiichiro Sumita, Graham Neubig, and Satoshi Nakamura. Guiding neural machine translation with retrieved translation pieces. In NAACL, 2018.
>
> [3] M. Xia, Guoping Huang, Lemao Liu, and Shuming Shi. Graph based translation memory for neural machine translation. In AAAI, 2019.
>
> [4] Deng Cai, Yan Wang, Huayang Li, Wai Lam, and Lemao Liu. Neural machine translation with monolingual translation memory. In ACL, 2021.

---

> > ### Comment · Reviewer_n9iv · 2022-11-28
> > **Thanks for the response.**
> >
> > Thanks for the very detailed and thoughtful responses to my notes! My primary concerns were the missing connection to Gu et al. (2018), the decoding speed and scalability claims, and clarity. The responses and edits include discussion for each of these points:
> > - they include a note in S3.3 highlighting the close connection to Gu et al. (2018).
> > - they include experiments on wmt to show decoding speeds with larger faiss indices—presumably resource constraints mean that this precludes the use of faiss-gpu. Regardless of whether these results transfer to other infra setups, it seems like a useful enough data point, and in the context of this study, the response makes a good effort to address decoding speed concerns.
> > - they include experiments on the size of the wmt corpus to address the scalability question.
> > - they add more detail for specific sections/experiments and note that code will be released.
> >
> > It’s a shame that some of these points could only be included in the appendix, but re-organizing the entire paper is more of a matter of preference.
> >
> > In light of the response and edits, I’ll raise my score to a 6.

---

> > > ### Author Response · Authors · 2022-11-29
> > > **Thanks for your reply**
> > >
> > > Thank you very much for your kind reply. Due to the limited time, we were only able to append our additional experiments to the supplementary content. In the future version, we will follow your suggestions move several important points into the main body of our paper.

---

### Official Review · Reviewer_wAt2 · 2022-11-03

**Confidence:** 3
**Correctness:** 3
**Technical Novelty And Significance:** 3
**Empirical Novelty And Significance:** 3
**Recommendation:** 6

**Clarity, Quality, Novelty And Reproducibility:**

The paper is clear, and the quality is good. The proposed idea is reasonable and novel but simple.

**Strength And Weaknesses:**

Strength: The proposed method is reasonable and effective to improve the kNN-MT framework by reducing the search space.
Weakness: More analysis on m and k choice is needed. If k=1 or 2, is the always best choice for kNN-MT, which means the rest examples are all far away, then we just keep top-1 token pair is ok and does not need sentence retrievel.

**Summary Of The Paper:**

This work is to improve the efficiency of the KNN-MT framework. In this work, the author proposed a simple and scalable framework to speed up the KNN search. By utilizing sentence level retrieval approach (BM25), the author reduced the search pool for each input. The experiment results shows comparable performance with KNN-MT and reduced time/storage cost.

**Summary Of The Review:**

Overall, the work is simple but effective but needs further justification mentioned in the k setup to verify if the method is solid or it is due to the task dataset.

---

> ### Author Response · Authors · 2022-11-19
> **Response to Reviewer wAt2**
>
> We would like to thank you very much for the detailed feedback and valuable suggestions! We will follow the suggestions on experiments and writing and revise our paper accordingly.
>
> **Q1: More analysis on k and m.**
>
> A1: As reported in previous papers [1,2], the optimal hyper-parameter k for kNN-MT is not large (8/4/16/4 for IT/Medical/Koran/Laws domain). When we construct a much smaller datastore dynamically, the hyper-parameter k should be undersized too. This is because the dynamic datastore we build does not contain much redundancy like the original datastore in kNN-MT, so arbitrarily introducing a large k will bring extra noise to SK-MT. The results of our hyper-parameter selection experiment in Section 4.1, Paragraph "The Effect of Hyper-parameters", reveal that k=1,2 is best for SK-MT. We also find that k=1,2 is best for the JRC-Acquis dataset, and the results are reported in Table 13 in our next draft. The number of reference samples (m) balances the performance and latency, which is discussed in Section 4.1, Paragraph "Decoding Speed and Storage Overhead." We also discuss the corpus scale's effect on m in Appendix A.5 of our new draft.
>
>
> References:
>
> [1] Urvashi Khandelwal, Angela Fan, Dan Jurafsky, Luke Zettlemoyer, and Mike Lewis. Nearest neighbor machine translation. In ICLR 2021
>
> [2] Xin Zheng, Zhirui Zhang, Junliang Guo, Shujian Huang, Boxing Chen, Weihua Luo, and JiajunChen. Adaptive nearest neighbor machine translation. In ACL 2021

---

### Decision · Program_Chairs · 2023-01-20

**Decision:**

Accept: poster

**Justification For Why Not Higher Score:**

The technical contribution is somewhat limited, and the paper doesn't evaluated the scalability on larger corpora.

**Justification For Why Not Lower Score:**

The proposed method is simple and scalable as advertised, and achieves strong empirical results, so I think it is a useful contribution to the community.

**Metareview: Summary, Strengths And Weaknesses:**

The submission introduces a method to improve the efficiency of kNN-MT, by first retrieving a set of relevant translation pairs with BM25 and using these as a kNN datastore. The authors also introduce a simple but effective technique for dynamically choosing interpolation weights.

The method is well motivated, and makes kNN-MT dramatically more practical by improving efficiency. The method for adaptive interpolation weights also simplifies previous work. The results are strong, and match or outperform the original kNN-MT.

One weakness is that there are not larger scale experiments on WMT19, which leaves it uncertain how well the method will scale. The authors note in their response that the baseline kNN-MT is too resource-intensive to re-implement in this setting, although a reviewer notes that they have re-implemented it on CPU. The technical contribution is also a bit limited compared to [1] and [2].

[1] Urvashi Khandelwal, Angela Fan, Dan Jurafsky, Luke Zettlemoyer, and Mike Lewis. Nearest neighbor machine translation. In ICLR 2021
[2] Jiatao Gu, Yong Wang, Kyunghyun Cho, and Victor O. K. Li. Search engine guided neural machine translation. In AAAI, 2018.



**Note From Pc:**

if the above contains the word "oral" or "spotlight" please see: "oral" presentation means -> notable-top-5% and "spotlight" means -> notable-top-25%. As stated in our emails, we are disassociating presentation type from AC recommendations

**Summary Of Ac-Reviewer Meeting:**

n/a